# Spatial clustering and temporal trend analysis of international migrants diagnosed with tuberculosis in Brazil

**Ricardo Alexandre Arcêncio**[1]*, **Thaís Zamboni Berra**[1], **Nahari de Faria Marcos Terena**[2], **Matheus Piumbini Rocha**[3], **Tatiana Ferraz de Araújo Alecrim**[4], **Fernanda Miye de Souza Kihara**[3], **Keila Cristina Mascarello**[5], **Carolina Maia Martins Sales**[3], **Ethel Leonor Noia Maciel**[3]

**1** Department of Maternal-Infant and Public Health Nursing, University of São Paulo at Ribeirão Preto School of Nursing, Ribeirão Preto, São Paulo, Brazil, **2** Department of Statistics, University Roma La Sapienza, Rome, Italy, **3** Epidemiology Laboratory, Health Sciences Center, Universidade Federal do Espírito Santo, Vitoria, Brazil, **4** Department of Nursing, Federal University of São Carlos, São Paulo, Brazil, **5** Department of Health Sciences, Centro Universitário Norte do Espírito Santo, Universidade Federal do Espírito Santo, Vitoria, Brazil

* ricardo@eerp.usp.br

**Data Availability Statement:** The data used in the present study were obtained through DATASUS, a Brazilian government website, which unifies the

## Abstract

### Background

Tuberculosis (TB) in migrants is of concern to health authorities worldwide and is even more critical in Brazil, considering the country´s size and long land borders. The aim of the study was to identify critical areas in Brazil for migrants diagnosed with TB and to describe the temporal trend in this phenomenon in recent years.

### Methods

This is an ecological study that used spatial analysis and time series analysis. As the study population, all cases of migrants diagnosed with TB from 2014 to 2019 were included, and Brazilian municipalities were considered as the unit of ecological analysis. The Getis-Ord Gi* technique was applied to identify critical areas, and based on the identified clusters, seasonal-trend decomposition based on loess (STL) and Prais-Winsten autoregression were used, respectively, to trace and classify temporal trend in the analyzed series. In addition, several municipal socioeconomic indicators were selected to verify the association between the identified clusters and social vulnerability.

### Results

2,471 TB cases were reported in migrants. Gi* analysis showed that areas with spatial association with TB in immigrants coincide with critical areas for TB in the general population (coast of the Southeast and North regions). Four TB clusters were identified in immigrants in the states of Amazonas, Roraima, São Paulo, and Rio de Janeiro, with an upward trend in most of these clusters. The temporal trend in TB in immigrants was classified as increasing in Brazil (+ 60.66% per year [95% CI: 27.21–91.85]) and in the clusters in the states of

data reported in the Notifiable Diseases Information System (SINAN) and can be accessed through this link: http://tabnet.datasus.gov.br/cgi/tabcgi.exe?sinannet/cnv/tubercbr.def To obtain the data, we put the following information: Line: notification municipality; Spine: Year of diagnosis; Period: 2014 to 2019. In the available selections, we include only immigrants.

**Funding:** ELNM received funding from the Pan American Health Organization (PAHO) [award number: 67278].

**Competing interests:** The authors have declared that no competing interests exist.

Amazonas, Roraima, and Rio de Janeiro (+1.01, +2.15, and + 2.90% per year, respectively). The cluster in the state of São Paulo was the only one classified as stationary. The descriptive data on the municipalities belonging to the clusters showed evidence of the association between TB incidence and conditions of social vulnerability.

## Conclusions

The study revealed the critical situation of TB among migrants in the country. Based on the findings, health authorities might focus on actions in regions identified, stablishing an intensive monitoring and following up, ensuring that these cases concluded their treatment and avoiding that they could spread the disease to the other regions or scenarios. The population of migrants are very dynamic, therefore strategies for following up them across Brazil are really urgent to manage the tuberculosis among international migrants in an efficient and proper way.

## Background

Tuberculosis (TB) is one of the diseases with greatest global magnitude and transcendence in public health, affecting millions of people a year and classified as the most lethal infectious disease in the world. According to estimates, in 2019 there were approximately ten million new TB cases in the world and 1.4 million TB deaths. In Brazil alone, some 74,000 new cases were diagnosed in the same year [1].

Most TB deaths occur in developing countries, including Brazil. The main explanation for this reality is social inequality in these countries, where many people are not diagnosed correctly and/or sufficiently early in health services, besides difficulties in obtaining appropriate treatment [1].

Evidence suggests that the pandemic caused by SARS-CoV-2 (severe acute respiratory syndrome coronavirus-2), responsible for COVID-19 (coronavirus disease), may have intensified this already adverse epidemiological scenario with TB. Many services have closed and stopped conducting their activities during the pandemic, such as directly observed treatment (DOT), where a healthcare worker assists the individual with TB in taking the medication as prescribed, according to World Health Organization (WHO) guidelines [2, 3].

Before the COVID-19 pandemic, global TB incidence had been decreasing, as reported by the WHO [1]. However, certain population groups are considered more vulnerable, with a trend towards increasing rates of the disease. These include homeless people, the prison population, indigenous people, slumdwellers, and refugees or migrating individuals. The latter have left their homelands (usually regions with high TB endemicity) due to civil unrest or in search of better living conditions, migrating to other territories, which may contribute to the difficulty in eliminating TB, as already reported by several authors [2–5].

TB infection and illness can also occur in the new contexts in which migrants live (rather than originating in their home countries), due to the high burden of the disease in countries that receive immigrants and the social vulnerability of these people when they arrive in the new country and need to resettle. They often occupy precarious residential areas with low coverage of essential services such as health, education, basic sanitation, and housing, the situation for many migrants in Brazil today. Several studies have shown the impact of migration on the resurgence of TB in Europe [2, 6–11]; however, in Brazil there is a lack of studies on this phenomenon, representing a serious knowledge gap.

A study in Brazil with a migrant population identified gender inequality, ethnic and racial prejudice, residential segregation, and low income as the main factors favoring TB infection, illness, and spread [12]. Brazil is an important destination for migrants, and migratory waves have grown significantly in recent years, especially with migrants searching for work and better opportunities and living conditions. Migrations are also motivated by political warfare, civil unrest, and religious persecution in the migrants´ home countries [12].

The current study thus aimed to identify critical areas in Brazil for migrants diagnosed with TB and describe the temporal trend of this phenomenon in recent years. Various methodological designs can achieve this aim, but few with a spatial approach associated with the analysis of time series; such studies can provide important evidence to back the elaboration of public policies and strategic actions to promote equitable access by these populations and assistance in controlling the disease.

## Methods

### Study scenario

This was an ecological study [13] in which the units of analysis were all 5,570 Brazilian municipalities (counties). Brazil, located in South America, is divided into five major geographic regions (North, Northeast, Central-West, Southeast, and South), with 26 states (federative units) plus the Federal District (DF). The country has a territory of 8,550,767 km$^2$ and an estimated population of 208.4 million [14].

According to the most recent annual report published by the Observatory of International Migrations (OBMigra) in 2019 [15], from 2011 to 2018 there were more than 774 thousand legally recorded migrants registered in Brazil, of whom approximately 492 thousand were recorded as long-term migrants, that is, immigrants who had stayed in the country for more than a year.

The migratory flows mostly involve young males and young people with secondary and university education. The occupational groups for most employed migrants in Brazil are in the production of industrial goods and services and sales in shops and markets. The main economic sectors are industry, commerce, and maintenance and repairs [15].

The leading nationalities of migrants in Brazil are Haitian, followed by Bolivian, Venezuelan, and Colombian. The regions with the most migrants are the Southeast (55.1%), especially the states of São Paulo (SP) (41.2%) and Rio de Janeiro (RJ) (9.4%), the South (20.5%), with the migration distributed evenly across its three states, namely Rio Grande do Sul (RS), Santa Catarina (SC), and Paraná (PR), and the North (8.6%), especially the state of Roraima (RR) (4.3%) due to migration from Venezuela [15].

### Study population

The study included all TB cases in immigrants reported from 2014 to 2019 notified in the Notifiable Diseases Information System (SINAN). SINAN is the Brazilian information system responsible for recording and processing information on diseases of mandatory notification throughout the country, as is the case of TB, providing bulletins and epidemiological reports, serving as one of the main surveillance systems in Brazil.

The system was updated in 2015 to include fields for specific populations such as prison inmates, homeless, healthcare workers, migrants, and other specific groups. Based on the migrant versus non-migrant variable, it is possible to identify case reports of migrants with TB.

As inclusion and exclusion criteria, cases were selected in which the notification form contained the date of diagnosis; cases in which the date of diagnosis was not recorded were

excluded from the analysis. The analysis also excluded cases in which the field for the munici-pality of notification was not completed.

## Analysis plan

**Spatial association analysis.** Initially, to verify the distribution of TB cases in migrants across Brazil, cases were grouped according to year of notification, and thematic maps were prepared using ArcGis software version 10.5 in order to facilitate the data´s interpretation.

Then, to identify the spatial association of TB, the technique called Getis-Ord Gi* was used, which allows verifying the association of TB locally, that is, considering the number of cases in each unit of analysis, based on a matrix of neighbors, that is, considering the mean number of events in neighboring municipalities [16].

This analysis generates a *z-score* with its respective p-values for each of the municipalities. The higher the *z-score*, the more intense the clustering of high values (hotspots), while the lower the *z-score*, the more intense the clustering of low values or the lower the event's occur-rence (coldspots). Besides the *z-score*, the analysis also furnishes the confidence interval (CI) for each of the clusters identified, assuming values for 99%, 95%, or 90% CI [16].

Four types of input values (or target variables) can be used for the analysis: counts (number of cases), rates (e.g., the proportion of the population with university degrees), averages (e.g., median or mean household income), and indices (e.g., a score indicating whether household spending on sporting goods is above or below the national average). For this specific study, the target variable was an event with a finite number of cases, it was decided to use as input value the number of cases of migrants diagnosed with TB according to Brazilian municipality and the number of TB cases in the general population [17], aimed at comparing the identified regions. ArcGis version 10.5 was used to produce the analysis and thematic map.

**Time series analysis.** To verify TB patterns in migrants in Brazil and in each cluster iden-tified in the Getis-Ord Gi* analysis during the study period and their temporal trends, sea-sonal-trend decomposition by loess (STL) was used, a method based on a locally weighted regression [18], where cases were grouped by month of notification, when available, and the number of monthly cases was smoothed by first-order moving averages as proposed by Beck-etti [19], in order to decrease the data's amplitude. RStudio software was used for this analysis and construction of the graphs.

Prais-Winsten autoregression was then used to classify the temporal trend of TB in migrants in Brazil and in the clusters as increasing, decreasing, or stationary in the study period. When the time trend was classified as increasing or decreasing, the monthly percent change (MPC) and respective 95%CI were calculated [20]. This stage was performed with STATA, version 14.

**Descriptive analysis.** The municipality´s characteristics, among other conditions, can influence the likelihood of an individual becoming infected and developing TB. Table 1 lists some of the factors that aggravate the spread of TB, suggesting an association between TB inci-dence and socioeconomic variables. The current study considered the variables income, sani-tation, primary healthcare, proportion of elderly people, and crude birth rate as indicators, all influenced by the population´s structure and development, to enable comparison of munici-palities identified in the clusters.

## Ethics approval and consent to participate

In compliance with Resolution 499/2012 of the National Health Council, the study was approved by the Institutional Review Board of the São Paulo Municipal Health Secretariat under Certificate of Ethical Review number 18703119.3.3002.0086 issued on July 24, 2020.

**Table 1. Specification of socioeconomic and demographic variables.**

| Variable | Definition of variable | Source | Period |
|---|---|---|---|
| MHDI | Municipal Human Development Index | Brazilian Institute of Geography and Statistics (IBGE) | 2010 |
| Social exclusion index | Corresponds to poverty rate. | IBGE | 2003 |
| GDP | per capita Gross Domestic Product. | IBGE | 2017 |
| Percentage of Primary Care coverage | Percentage of population assisted by primary healthcare. | E-Gestor Atenção Básica | 2019 |
| Population size | Total population of the municipality. | IBGE | 2010 |
| Proportion of elderly | Proportion of elderly in the municipal population. | IBGE | 2010 |
| Crude birth rate | Number of live births per 1,000 inhabitants in the municipality. | DATASUS | 2008 |
| TB incidence rate | Number of new reported TB cases per 1,000 inhabitants in the municipality. | DATASUS | 2019 |
| Healthcare worker rate | Total number of healthcare workers per 1,000 inhabitants in the municipality. | National Registry of Healthcare Establishments (CNES) | 2009 |
| Basic sanitation rate | Proportion of households with basic sanitation. | DATASUS and IBGE | 2000 |

Consent to participate was not applicable, because the work was performed using secondary data from cases of migrants diagnosed with TB and recorded in the SINAN database. The database used for the study is anonymized, so it is not possible to identify the subjects included in the study. The database was accessed from September to November 2020.

## Results

From 2014 to 2019, 539,708 TB cases were reported in Brazil, with 7,499 cases reported in migrants (1.4%). Of these, 2,466 cases (32.8%) met the inclusion criteria and were included in the study.

Of the 2,466 TB cases identified in migrants and included in the study, 59% were males. Thirty percent of the migrant women diagnosed with TB were pregnant at the time of diagnosis. As for age, most cases were individuals in the economically active bracket, with 15% of cases between 25 and 29 years of age and 9% in the elderly population (over 60 years). About 65% of TB notifications in migrants were in individuals self-identified as black or brown and 6% self-declared indigenous (compared to about 1% in the non-migrant population). Approximately 12% of migrants diagnosed with TB reported that they were HIV-positive.

Fig 1 shows the distribution of TB cases in migrants over time, with an increase in cases, especially in the South and Southeast of Brazil.

The Getis-Ord Gi* technique showed that areas with spatial association for TB in migrants coincide with areas identified for TB in the general population, and that the most affected areas are the Southeast coastline and the area in and around the state of Amazonas in the North.

Fig 2 shows the areas with high and low TB clusters in the general population and TB in migrants in Brazil. For the general population, a coldspot cluster was identified in the South of Brazil, while a hotspot cluster for TB in migrants was found in the state of Roraima (RR) along the border between Brazil and Venezuela. The analysis considered all cases notified in Brazil in the study period, for TB cases in both the general population and in migrants, since the Getis-Ord Gi* does not include time as a variable.

Four TB clusters were identified in migrants by the Getis-Ord Gi* analysis, all of which classified as high clusters (Fig 3). One cluster was identified in the North of Brazil in the state of Amazonas, consisting of seven municipalities, namely Careiro da Várzea, Iranduba, Itacoatiara, Manaus (state capital), Novo Airão, Presidente Figueiredo, and Rio Preto da Eva.

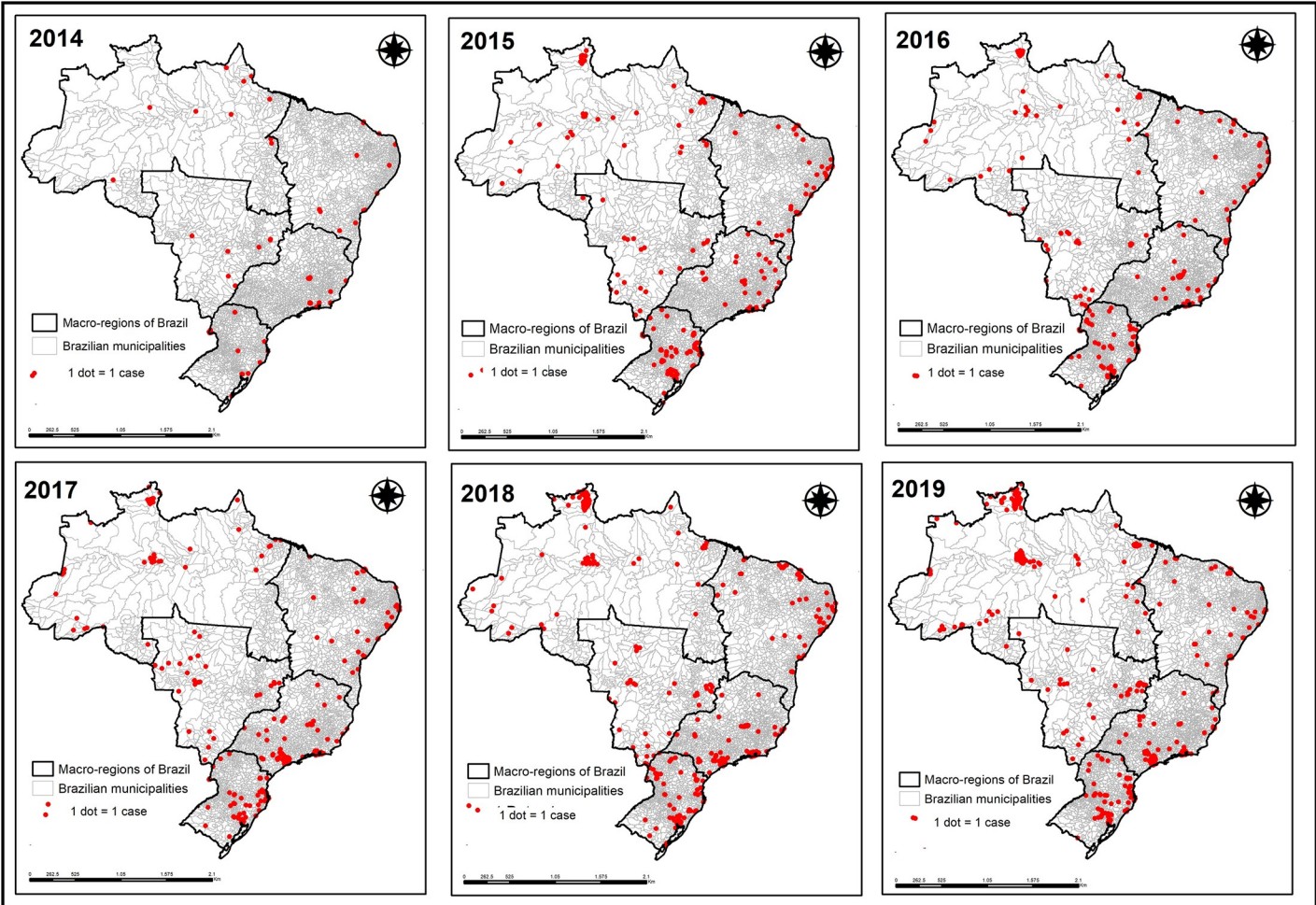

**Fig 1. Distribution of tuberculosis cases in migrants according to year of notification, Brazil (2014–2019).** Source: prepared by the authors.

Another cluster was identified in the North of Brazil in the state of Roraima, consisting of nine municipalities: Amajari, Alto Alegre, Boa Vista (state capital), Bonfim, Cantá, Mucajaí, Normandia, Pacaraima, and Uiramutã.

Two clusters were identified in Southeast Brazil, one of which in the state of São Paulo and consisting of 25 municipalities (São Vicente, Taboão da Serra, Arujá, Barueri, Caieiras, Cajamar, Cotia, Diadema, Embu, Embu-Guaçu, Ferraz de Vasconcelos, Guarulhos, Itanhaém, Itapecerica da Serra, Itaquaquecetuba, Juquitiba, Mariporã, Mauá, Osasco, Poá, Santana de Parnaíba, Santo André, São Bernardo do Campo, São Caetano do Sul, and the state capital São Paulo) and the other in the state of Rio de Janeiro, consisting of eight municipalities (Duque de Caxias, Nilópolis, Nova Iguaçu, São Joaquim de Meriti, Seropédica, Mesquita, Itaguaí, and the state capital Rio de Janeiro).

For the time series analysis, we included again the 2,466 TB cases in migrants that had the date of diagnosis on the notification form. As shown in Fig 4, the time trend for TB in migrants in Brazil is increasing, while for the time series in the clusters identified in the states of Amazonas and Roraima, both of which in the North of Brazil, the temporal trend was stationary until 2018, when it began an upward trend.

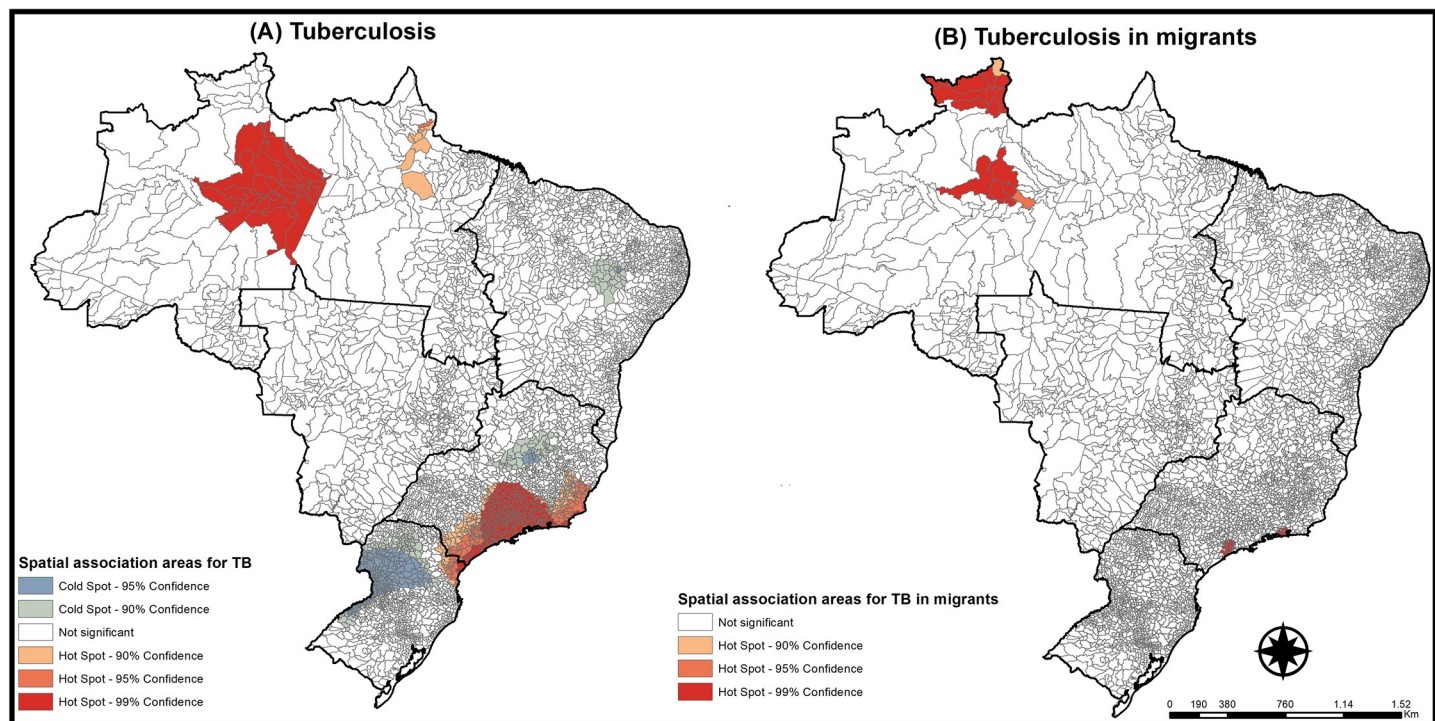

**Fig 2. High and low tuberculosis clusters in the general population and in migrants, Brazil (2014–2019).** Source: prepared by the authors.

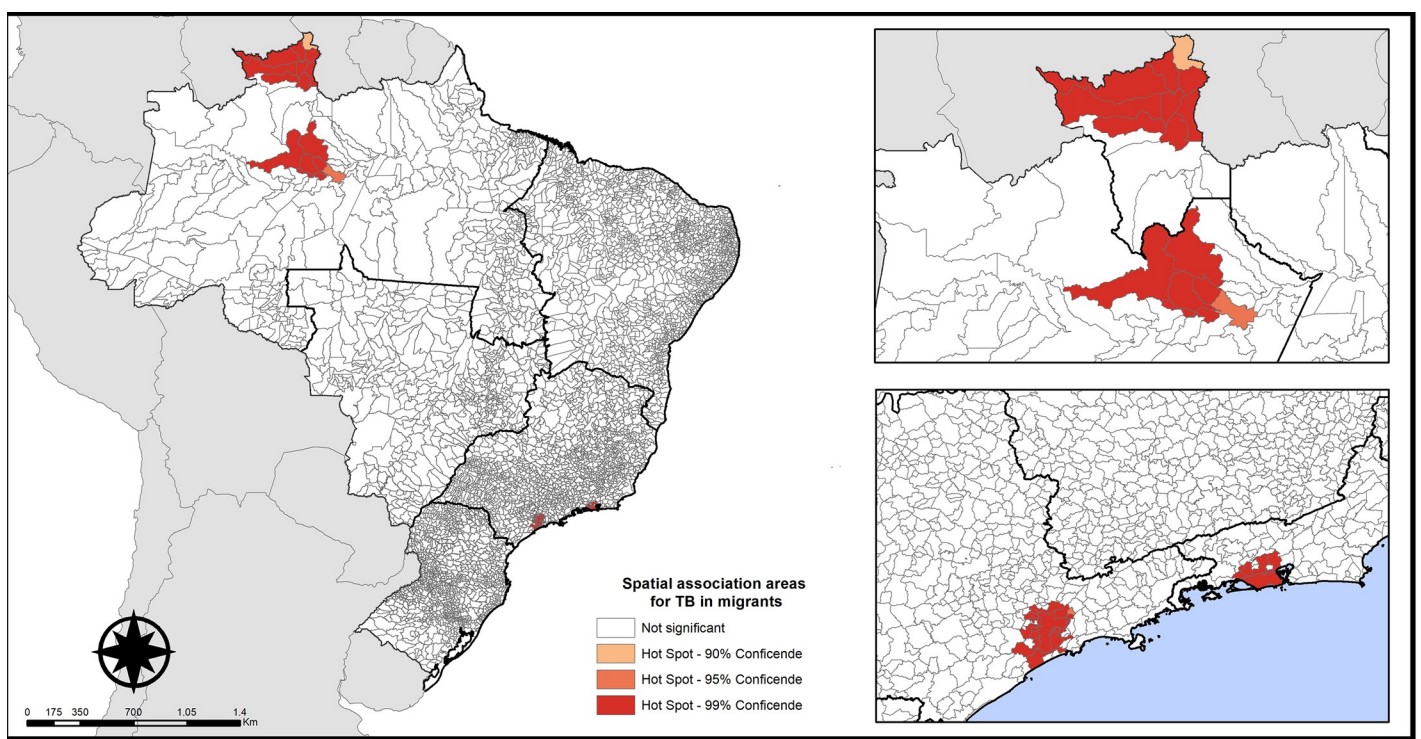

**Fig 3. Spatial association areas for tuberculosis in migrants, Brazil (2014–2019).** Source: prepared by the authors.

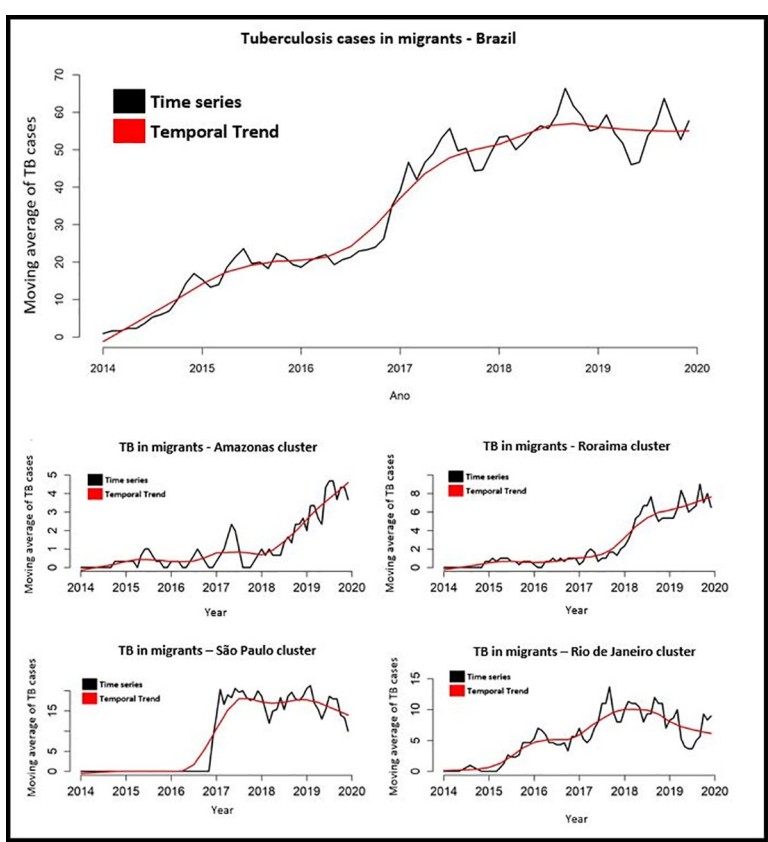

**Fig 4. Time series and temporal trend for tuberculosis in migrants in Brazil and in the clusters identified by Getis-Ord Gi\* analysis, Brazil (2014–2019).** Source: prepared by the authors.

Meanwhile, the time series in the cluster in the state of São Paulo (Southeast Brazil) does not show any cases from 2014 to 2017, when the time trend displayed a stationary trend. Finally, the cluster identified in the state of Rio de Janeiro, also in Southeast Brazil, showed an upward trend until 2018, when it turned downward.

Table 2 presents the results of the Prais-Winsten analysis, confirming the findings in Fig 4. The TB temporal trend in migrants in Brazil was classified as increasing, with a 60.66% monthly increase (95%CI: 27.21–91.85), as were the clusters in the states of Amazonas, Roraima, and Rio de Janeiro, which also had their temporal trends classified as increasing (with monthly increases of 1.01, 2.15, and 2.90%, respectively). Finally, the temporal trend in the cluster in the state of São Paulo was classified as stationary. The clusters that were identified represent about 60% of cases reported in Brazil during the study period.

**Table 2. Tuberculosis temporal trend in migrants in Brazil and in clusters identified by Getis-Ord Gi\* analysis, Brazil (2014–2019).**

| Location | Coefficient ($CI_{95\%}$) | Trend | MPC ($CI_{95\%}$) (95%CI) |
|---|---|---|---|
| Brazil | 0.85 (0.63–1.08) | Upward | 50.66% (27.21–91.85) |
| Cluster–Amazonas | 0.05 (0.02–0.07) | Upward | 1.01 (0.39–1.45) |
| Cluster–Roraima | 0.10 (0.06–0.14) | Upward | 2.15 (1.23–3.16) |
| Cluster–São Paulo | 0.21 (-0.01–0.44) | Stationary | - |
| Cluster–Rio de Janeiro | 0.13 (0.05–0.21) | Upward | 2.90 (1.01–5.18) |

As shown in Table 3, among the municipalities belonging to the clusters identified by Getis-Ord Gi*, the city of São Caetano do Sul (São Paulo state) had the highest HDI (0.862), higher than the state average (0.770), while the lowest HDI (0.453) was in Uiramutã (Roraima), lower than the state average of 0.598. Regarding Gross Domestic Product (GDP), the lowest index (10.73) was observed in the municipality of Careiro da Várzea (Amazonas), below the state average of 15.96, while the highest index (82.12) was observed in São Caetano do Sul (São Paulo), almost double the state average (45.31).

The lowest crude birth rate was in the municipality of Careiro da Várzea (Amazonas) (6.6), nearly three times below the state average (18.3), and the highest birth rate was in Uiramutã (51.8), while the state average in Roraima was 32.4. The municipality of Uiramutã also had the lowest proportion of elderly people in its population (3.7), while the average for the state of Roraima was 5.6; the highest proportion of elderly people was in Cajamar (São Paulo) (18.8), above the state average of 15.7.

Regarding health indicators, the municipality with the lowest primary healthcare coverage was Cotia (São Paulo) with 28.79%, below the state coverage rate (55.6%). Fourteen municipalities had 100% PHC coverage, six of which in the state of Amazonas (Careiro da Varzea, Iranduba, Itacoatiara, Novo Airão, Presidente Figueiredo, and Rio Preto da Eva), one in Rio de Janeiro (Nilópolis), and seven in Roraima (Amajari, Alto Alegre, Bonfim, Cantá, Mucajai, Pacaraima, all Uiramutã), all above the state averages (93.09%, 66.64%, and 94.52% respectively, in the states of Amazonas, Rio de Janeiro, and Roraima).

As for healthcare worker rate, the municipality with the lowest rate was Cantá (Roraima) (0.6), below the state average of 3.1, while the municipality with the highest rate was São Caetano do Sul (São Paulo) (8.6), nearly three times the state average of 3.2.

The municipality with the lowest level of basic sanitation was Uiramutã (Roraima), with zero, that is, there is no basic sanitation in this municipality, followed by Alto Alegre, also in Roraima, with an index of 0.1, below the already extremely low state average of 2.05. The municipality of Careiro da Várzea (Amazonas) also had a basic sanitation index of 0.1, below the state average of 7.85. The municipality with the highest level of basic sanitation was São Caetano do Sul (São Paulo) with 99.4, while the state average was 60.48.

The lowest and highest social exclusion indices were observed in municipalities in the state of São Paulo, which had an average of 0.4 for this index. The lowest index was in São Caetano do Sul (0.1), and the highest was in Itaquaquecetuba (0.7), above the state average, showing great social disparities within the same state.

Finally, the municipality with the lowest TB incidence rate was Cajamar (São Paulo) with 0.1 cases/1,000 inhabitants, while the municipality with the highest rates was São Vicente, a coastal city in São Paulo state (1.7/1,000 inhabitants). The average state TB incidence rate was 0.5/1,000 inhabitants.

## Discussion

This study aimed to identify critical areas in Brazil for TB in migrants and to describe this phenomenon´s temporal trend in recent years. The results allowed identifying four clusters with spatial association for migrants diagnosed with TB. Thus, most TB cases in migrants were grouped in municipalities in the states of Roraima, Amazonas, São Paulo, and Rio de Janeiro. The findings also evidenced the temporal trend of reported TB cases in migrants as increasing, both in Brazil and in the clusters identified in the spatial analysis, with the exception of São Paulo, which showed a stationary temporal trend for TB cases in migrants.

As for migration, most Brazilian states receive people from other countries, but the highest concentration of migrants occurs in some states such as Rio de Janeiro and São Paulo, where

**Table 3. Characteristics of social and health determinants in the municipalities with tuberculosis clusters in migrants, Brazil (2014–2019).**

| | HDI | Social exclusion index | GDP | Population size | Crude birth rate | Proportion of elderly | Percentage of Primary Care coverage | Healthcare worker rate | Basic sanitation rate | TB incidence rate |
|---|---|---|---|---|---|---|---|---|---|---|
| Amazonas | | | | | | | | | | |
| Manaus | 0.737 | 0.4 | 34,363 | 1,802,014 | 22.4 | 6.0 | 51.68 | 3.0 | 32.2 | 1.6 |
| Careiro da Várzea | 0.568 | 0.2 | 10,730 | 23,930 | 6.6 | 7.8 | 100.00 | 0.9 | 0.1 | 0.2 |
| Iranduba | 0.613 | 0.6 | 14,858 | 40,781 | 23.8 | 7.0 | 100.00 | 2.6 | 0.5 | 1.4 |
| Itacoatiara | 0.644 | 0.6 | 19,817 | 86,839 | 23.8 | 7.3 | 100.00 | 2.0 | 1 | 0.7 |
| Novo Airão | 0.570 | 0.6 | 7,077 | 14,723 | 19.4 | 6.4 | 100.00 | 1.0 | 1.6 | 1.0 |
| Presidente Figueiredo | 0.647 | 0.5 | 12,866 | 27,175 | 19 | 5.2 | 100.00 | 2.2 | 19.4 | 0.9 |
| Rio Preto da Eva | 0.611 | 0.6 | 11,994 | 25,719 | 13.6 | 5.3 | 100.00 | 2.4 | 0.2 | 0.5 |
| Rio de Janeiro | | | | | | | | | | |
| Rio de Janeiro | 0.799 | 0.2 | 51,776 | 6,320,446 | 13.4 | 14.8 | 50.5 | 5.2 | 76.3 | 1.3 |
| Duque de Caxias | 0.711 | 0.5 | 45,895 | 855,048 | 15.3 | 10.0 | 40.76 | 3.9 | 55.3 | 0.9 |
| Nilópolis | 0.753 | 0.3 | 16,699 | 157,425 | 13.6 | 13.2 | 100 | 3.3 | 79 | 0.8 |
| Nova Iguaçu | 0.713 | 0.5 | 21,078 | 796,257 | 13.4 | 10.6 | 68.11 | 2.5 | 50.3 | 1.0 |
| São João de Meriti | 0.719 | 0.5 | 19,968 | 458,673 | 14 | 11.3 | 61.3 | 1.1 | 66.3 | 0.9 |
| Seropédica | 0.713 | 0.5 | 49,882 | 78,186 | 14.5 | 10.0 | 91.3 | 1.6 | 11 | 0.6 |
| Mesquita | 0.737 | n/a | 13,505 | 168,376 | 13 | 11.5 | 77.85 | 1.8 | n/a | 1.1 |
| Itaguaí | 0.715 | 0.5 | 61,820 | 109,091 | 17.1 | 9.4 | 43.36 | 5.4 | 39.6 | 0.9 |
| Roraima | | | | | | | | | | |
| Boa Vista | 0.752 | 0.4 | 26,924 | 284,313 | 23.5 | 5.2 | 57.05 | 4.1 | 14.6 | 0.8 |
| Amajari | 0.484 | 0.4 | 14,119 | 9,327 | 35.6 | 5.9 | 100 | 1.1 | 0.6 | 1.7 |
| Alto Alegre | 0.542 | 0.2 | 17,453 | 16,448 | 33.8 | 6.0 | 100 | 1.9 | 0.1 | 1.1 |
| Bonfim | 0.626 | 0.5 | 22,360 | 10,943 | 28 | 5.9 | 100 | 1.9 | 0.2 | 0.8 |
| Cantá | 0.619 | 0.4 | 14,685 | 13,902 | 24 | 7.3 | 100 | 0.6 | 0.3 | 0.7 |
| Mucajaí | 0.665 | 0.5 | 17,804 | 14,792 | 24.7 | 7.0 | 100 | 1.4 | 0.8 | 0.3 |
| Normandia | 0.594 | 0.2 | 14,108 | 8,940 | 42.7 | 4.9 | 93.71 | 1.7 | 0.4 | 0.6 |
| Pacaraima | 0.650 | 0.5 | 13,881 | 10,433 | 28.3 | 4.6 | 100 | 2.2 | 1.5 | 1.4 |
| Uiramutã | 0.453 | 0.6 | 11,847 | 8,375 | 51.8 | 3.7 | 100 | 1.0 | 0 | 0.2 |
| São Paulo | | | | | | | | | | |
| São Paulo | 0.805 | 0.3 | 57,759 | 11,253,503 | 15.8 | 15.8 | 62.59 | 5.2 | 85.7 | 0.7 |
| Arujá | 0.784 | 0.5 | 61,459 | 74,905 | 15.3 | 18.2 | 62.98 | 2.1 | 26.2 | 0.5 |
| Barueri | 0.786 | 0.5 | 177,748 | 240,749 | 18.2 | 15.4 | 69.94 | 7.6 | 78.8 | 0.6 |
| Caieiras | 0.781 | 0.4 | 33,491 | 86,529 | 15.8 | 15.6 | 49.69 | 3.4 | 68.7 | 0.2 |
| Cajamar | 0.728 | 0.6 | 20,.963 | 64,114 | 18.6 | 18.8 | 73.3 | 3.4 | 64.8 | 0.1 |
| Cotia | 0.780 | 0.5 | 48,679 | 201,150 | 18.3 | 17.8 | 28.79 | 3.0 | 49.8 | 0.5 |
| Diadema | 0.757 | 0.4 | 32,098 | 386,089 | 17.1 | 15.9 | 84.99 | 4.3 | 92 | 0.5 |
| Embu das artes | 0.735 | 0.5 | 43,860 | 240,230 | 18.1 | 14.6 | 51.51 | 1.4 | 57.6 | 0.4 |
| Embu-Guaçu | 0.749 | 0.5 | 15,285 | 62,769 | 14.8 | 14.8 | 52.63 | 1.6 | 13.5 | 0.5 |
| Ferraz de Vasconcelos | 0.738 | 0.6 | 15,830 | 168,306 | 16.8 | 17.5 | 33.05 | 3.1 | 72.8 | 0.4 |
| Guarulhos | 0.763 | 0.4 | 41,319 | 1,221,979 | 16.2 | 16.4 | 41.67 | 3.7 | 74.6 | 0.5 |
| Itanhaém | 0.745 | 0.3 | 18,812 | 87,057 | 16.2 | 13.3 | 81.94 | 3.0 | 14.3 | 1.3 |
| Itapecerica da Serra | 0.742 | 0.5 | 19,201 | 152,614 | 18.1 | 15.0 | 49.51 | 3.4 | 24.3 | 0.5 |
| Itaquaquecetuba | 0.714 | 0.7 | 19,221 | 321,770 | 15.4 | 15.6 | 38.74 | 1.5 | 66.8 | 0.5 |
| Juquitiba | 0.709 | 0.5 | 15,653 | 28,737 | 14.3 | 15.2 | 77.32 | 1.3 | 10.2 | 0.4 |

*(Continued)*

**Table 3.** (Continued)

| | HDI | Social exclusion index | GDP | Population size | Crude birth rate | Proportion of elderly | Percentage of Primary Care coverage | Healthcare worker rate | Basic sanitation rate | TB incidence rate |
|---|---|---|---|---|---|---|---|---|---|---|
| Mairiporã | 0.788 | 0.4 | 17,957 | 80,956 | 15 | 14.2 | 45.44 | 2.2 | 25 | 0.3 |
| Mauá | 0.766 | 0.5 | 35,252 | 417,064 | 14.5 | 14.2 | 40.09 | 2.5 | 74.5 | 0.4 |
| Osasco | 0.776 | 0.4 | 111,638 | 666,740 | 15.3 | 14.8 | 40.18 | 4.1 | 69.5 | 0.5 |
| Poá | 0.771 | 0.5 | 36,511 | 106,013 | 15.8 | 16.3 | 44.67 | 1.4 | 86.7 | 0.3 |
| Santana de Parnaíba | 0.814 | 0.4 | 65,083 | 108,813 | 14.5 | 14.8 | 56.59 | 1.9 | 33.4 | 0.2 |
| Santo André | 0.815 | 0.3 | 38,408 | 676,407 | 13.1 | 14.6 | 52.42 | 6.2 | 89.6 | 0.4 |
| São Bernardo do Campo | 0.805 | 0.3 | 53,999 | 765,463 | 14 | 14.4 | 63.6 | 3.3 | 85.1 | 0.4 |
| São Caetano do Sul | 0.862 | 0.1 | 82,120 | 149,263 | 11 | 16.7 | 97.52 | 8.6 | 99.4 | 0.2 |
| São Vicente | 0.768 | 0.2 | 14,441 | 332,445 | 15.9 | 16.6 | 33.48 | 1.5 | 64.6 | 1.7 |
| Taboão da Serra | 0.769 | 0.4 | 31,627 | 244,528 | 19.6 | 18.1 | 58.07 | 2.7 | 84.3 | 0.5 |

there is a higher concentration of TB cases in both the local population and the migrant population. The principal nationalities of migrants in these regions are Haitian, with a large presence in the work market, followed by Venezuelan, Senegalese, Bolivian, Colombian, and Bengali. Most of the immigrants leave their home countries in search of better work opportunities and better living conditions [15].

Contrary to immigration in Brazil in the 20th century, consisting mostly of individuals from the global North, basically Europeans, in the 21st century this migratory profile has changed, and now most immigrants are from the global South. This aspect is important epidemiologically, since most of these countries are heavily affected by TB [1], so migration may contribute to the spread of the disease between these countries and thus make TB control more difficult. When analyzing migration from a gender perspective, it is important to note that most migrants are males, who tend to experience higher TB prevalence than females, as reported in the literature [21–24].

Fig 1 shows that from 2014 to 2019 there was a major increase in migration to the Northeast and Central-West regions of Brazil, but the main regions reporting migrants with TB are still the South, Southeast, and North.

According to an OBMigra report [15], the nations with the most migrants to Brazil in 2019 were Venezuela, Paraguay, Bolivia, and Haiti, representing 53% of the total records. Corroborating the data presented in Fig 1, according to the same report [15], from 2010 to 2019 the main regions of Brazil that received migrants were the Southeast (44% of all records), concentrated mainly in the states of São Paulo and Rio de Janeiro, the South, totaling 22% of the records, distributed equally across the region´s three states (Paraná, Santa Catarina, and Rio Grande do Sul), and the North, with 20% of all the records, concentrated mainly in the states of Roraima and Amazonas (the latter, mostly in municipalities close to the state capital, Manaus).

In 2019, among registered migrants in the North, Roraima represented 38% of all records and had the most annual records since 2010, an increase that resulted from Venezuelan immigration to the region [15].

Regarding the labor market, the total number of immigrants with formal work contracts increased from 55.1 thousand in 2010 to 116.4 thousand in 2014, and then to 147.7 thousand

in 2019, or an annual growth of nearly 10%, mostly consisting of Haitians and Venezuelans. In 2010, formal workers were mostly concentrated in the Southeast (mainly São Paulo), but over the years there was a decrease in concentration in this region, which was redistributed to the South and Central-West [15].

A key challenge for the Brazilian policy to control diseases such as TB is the country´s continental size, with one of the longest land borders in the world. Brazil´s land border represents 68% of the country's entire territorial extension. Brazil thus has direct contact with ten of the 12 countries that make up South America, the exceptions being Chile and Ecuador. Brazil borders to the north on Suriname, Guyana, Venezuela, and French Guiana, to the northwest on Colombia, to the west on Peru and Bolivia, to the southwest on Paraguay and Argentina, and to the south on Uruguay [25].

When observing the results regarding the most critical regions for migrants diagnosed with TB, it is important to note that these same regions already suffer from structural inequality in these municipalities. Many of these migrants confront this reality when they arrive in Brazil, remaining in conditions of inequality and poverty.

When analyzing the temporal trends, most of the clusters presented upward trends. However, São Paulo was classified as having a stationary temporal trend. Still, it important to carefully examine the abrupt change that has occurred in this state since 2017, which may reflect the migratory flow that has intensified since then, mainly from other Latin American countries. Another hypothesis is underreporting and/or underdiagnosis in this population from 2014 to 2017, since the city did not report any migrants diagnosed with TB during that period. According to data from OBMigra [15], São Paulo is the state of Brazil with the most extensive regularization of employment relations with migrants, based on formal work contracts. The state thus has more registered migrants, and more TB diagnoses would be expected in this population.

Although the temporal trend in the cluster in Rio de Janeiro was classified as increasing, the graph in Fig 4 shows a drop in the trend from mid-2018 to late 2019. It is important to elucidate what may have caused this drop, which may reflect underreporting, so further studies are recommended in the region to better understand the phenomenon.

Regarding the Getis-Ord Gi* technique used to identify clusters with spatial association for TB in migrants, many studies have used this methodology in public health through the global dissemination of the use of geographic information systems (GIS) [26–30]. The technique´s s use is relevant, as it allows observing spatial associations based on a neighborhood matrix and thus grouping high and low values for the target event.

The identification of these areas is of great importance for public health and epidemiological surveillance services, so that greater efforts are needed for areas identified as critical, which display clustering of high values for the target event. Thus, if no measures are taken to identify cases and institute the correct treatment, the number will tend to increase, as corroborated by the time series analysis, which identified upward trends in cases in most of the clusters. The spatial approach, combined with time series, essentially allows predicting the problem´s exacerbation in the coming years if timely corrective measures are not taken.

Based on the descriptive data from the municipalities belonging to the clusters, we compared their characteristics and found that even within the same state, there are large structural differences. When the comparison is made between municipalities from different states, this segregation appears to be much greater. The current study thus provides an indication of the association between variables related to social vulnerability and TB, which can occur in the main areas chosen by migrants to settle in the new country.

Studies have shown that violence, labor exploitation, and sexual harassment are frequently reported during migration and in the country of destination [31, 32]. Stressful, stigmatizing,

and vulnerable conditions thus contribute to the development of non-infectious and infectious diseases in this population, including TB infection or reactivation of latent TB infection (LTBI) [33–35].

Most migrants are from low and middle-income countries and are generally headed for large cities in the new country [35, 36]. These migrants are more likely to settle in low-income neighborhoods in the city center and tend to have higher TB burden. As a result, this population often experiences more poverty, vulnerability, and social exclusion than in their home communities [35, 37]. There has been a major increase in migrants to (and within) South America in the last 15 years, and although this growing number can contribute to greater social integration, it can also increase the social vulnerability of these individuals and TB cross-contamination, as already evidenced in studies [38].

In view of the above, countries receiving large numbers of migrants should be able to increase their institutional capacity in a qualified way, aiming to help migrants with strengthened health services and trained professionals to deal with the increase in migratory flows. Countries with higher TB burden should recognize the signs and symptoms of the disease and ensure the provision of quality healthcare. In this context, inter-sector TB control strategies are necessary to tackle social and health inequalities, highlighting that both the migrant population and vulnerable local groups must be served [35].

The study´s limitations feature its ecological design, so that its results should not be transposed and/or interpreted at the individual level (ecological fallacy). The data used in the current study are from secondary sources, which can cause a bias common to this type of study due to completion errors, missing data, or even non-homogeneous updating of data in Brazil´s 5,570 municipalities.

TB diagnosis in migrants is also subject to underdiagnosis and underreporting, mainly due to their difficulty in accessing health services. Despite these caveats, the study used official data sources, which supports the results´ reliability.

One difficulty in this study involved obtaining specific population data on the number of registered migrants per municipality in Brazil. No reliable source was found with these records, so we urge federal officials to feed the systems regularly, not only at the state level, but also at the municipal level.

This is a pioneering study on TB in migrants from the perspective of two different methods, spatial and time series analysis. We recommend further studies to better understand TB in migrants and improve the data quality, mainly for specific populations, allowing calculation of rates and potentially additional results. In addition, we highlight the possibility of crossing data from different sources, as we are talking about two issues that are heavily loaded by social stigmas, TB and migrants, so crossing between different data sources can minimize some biases.

Literature reviews [39–41] have shown that the issue of TB in migrants has received little attention in Brazilian research, and this study helps fill this critical knowledge gap. Future research should include on-site studies to identify the clinical and epidemiological characteristics of migrants and their degree of exposure and contacts, besides prevalence of latent TB. As highlighted above, the current study used data from notified cases, while it is known that there are linguistic, cultural, economic, geographic, and organizational barriers that prevent this population´s access to health services, so that further studies are encouraged through national surveys that address the theme in more depth, with active case searches in communities and screening of migrants.

Finally, although migration may not be the direct cause of illness, it can increase individual vulnerability, considering the differences between migrants and the local population, besides differences between health systems in the respective countries. All these characteristics should

be considered for designing and implementing non-discriminatory, ethically based public health policies so that migrants can receive efficient healthcare [42].

Immigrants face quality of life issues when they arrive in a new country, associated with the working and living conditions to which they are exposed. They can encounter work risks that impact their health and safety. According to studies [43, 44], migrant workers generally work in sectors sensitive to economic fluctuations and tend to perform more hazardous and less healthy jobs, with less stability and in less qualified occupations such as construction, domestic services (performed mainly by women), cleaning services, and vending.

It is thus of paramount importance that public health policies are developed according to the needs of migrant populations and that health professionals are prepared to serve this population, using accessible and non-discriminatory language.

## Conclusions

The End TB Strategy proposed by WHO recognizes migrants as one of the populations most vulnerable to illness from TB and stresses that they must be considered a priority in treatment. However, many countries do not have specific policies to combat TB in this specific population, which makes the topic highly relevant and current.

The use of two different approaches (spatial and time series analysis) allowed identifying critical areas in Brazil for TB in migrants, describing the temporal trend of this phenomenon in recent years. Based on the results, in the absence of inclusive measures and appropriate public policies, we conclude that the situation may become worse, thus requiring mobilization and political commitment to TB control, especially in these more vulnerable population groups.

The study used data from reported TB cases from 2014 to 2019, thus before the COVID-19 pandemic. Considering the impact of COVID-19 on health services, such as suspension of community activities and a decline in DOTS coverage, the situation with TB in migrants may have become worse since 2020, requiring even more attention from health authorities and civil society.

## Author Contributions

**Conceptualization:** Ricardo Alexandre Arcêncio, Carolina Maia Martins Sales, Ethel Leonor Noia Maciel.

**Data curation:** Ricardo Alexandre Arcêncio, Thaís Zamboni Berra, Nahari de Faria Marcos Terena, Matheus Piumbini Rocha, Tatiana Ferraz de Araújo Alecrim, Carolina Maia Martins Sales, Ethel Leonor Noia Maciel.

**Formal analysis:** Ricardo Alexandre Arcêncio, Thaís Zamboni Berra, Nahari de Faria Marcos Terena, Tatiana Ferraz de Araújo Alecrim, Carolina Maia Martins Sales, Ethel Leonor Noia Maciel.

**Funding acquisition:** Carolina Maia Martins Sales, Ethel Leonor Noia Maciel.

**Investigation:** Fernanda Miye de Souza Kihara, Keila Cristina Mascarello, Ethel Leonor Noia Maciel.

**Methodology:** Ricardo Alexandre Arcêncio, Thaís Zamboni Berra, Matheus Piumbini Rocha, Fernanda Miye de Souza Kihara, Keila Cristina Mascarello, Carolina Maia Martins Sales, Ethel Leonor Noia Maciel.

**Project administration:** Ricardo Alexandre Arcêncio.

**Resources:** Ricardo Alexandre Arcêncio.

**Supervision:** Ethel Leonor Noia Maciel.

**Writing – original draft:** Ricardo Alexandre Arcêncio, Thaís Zamboni Berra, Nahari de Faria Marcos Terena, Matheus Piumbini Rocha, Tatiana Ferraz de Araújo Alecrim, Fernanda Miye de Souza Kihara, Keila Cristina Mascarello, Carolina Maia Martins Sales, Ethel Leonor Noia Maciel.

**Writing – review & editing:** Ricardo Alexandre Arcêncio, Thaís Zamboni Berra, Nahari de Faria Marcos Terena, Matheus Piumbini Rocha, Tatiana Ferraz de Araújo Alecrim, Fernanda Miye de Souza Kihara, Keila Cristina Mascarello, Carolina Maia Martins Sales, Ethel Leonor Noia Maciel.

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
