## [Decision Letter · Decision Letter 0]

1 Mar 2021

PONE-D-20-40788

Spatial clustering and temporal trend analysis of international migrants diagnosed with tuberculosis in Brazil: Are we really managing to control the disease?

PLOS ONE

Dear Dr. Mascarello,

Thank you for submitting your manuscript to PLOS ONE. After careful consideration, we feel that it has merit but does not fully meet PLOS ONE’s publication criteria as it currently stands. Therefore, we invite you to submit a revised version of the manuscript that addresses the points raised during the review process.

This is an important and relevant paper. The paper makes an important contribution to discuss this neglect issue in Brazil, especially when considering migrants. However, based on the reviewers and my own reading I believe the paper should revised some points: 

please, revise the context. I think that the paper has data to analyze if there are more migrants with TB in the country and where are they, but is limited in the discussion on how the country is dealing with the issue;I share a concern with one the reviewers on the method. Please, the methods part could be more developed and indicate more details on the methods. Here are few reasons and doubts:spatial clusters assume that we have a continum of cases (in any variable of interest) - but this is not the case in the analysis since you are working with a very specific sub-population group. It would be important to clarify this question in the paper and also discuss possible limitations of the method in the studysimilarly, in recent years there was an increase of immigrans to Brazil. So, the time trend analysis should take that into consideration. How the increase in inflow might impact the analysis and observed trends? What are the alternatives to that? Not only that, but also the profile of migrants might have changed and they would also impact in the discussion of time trendsPlease, based on the analysis and discussion - the paper could improve the discussion of public health policy implications. Please, see also detailed comments by the reviewers. 

We look forward to receiving your revised manuscript.

Kind regards,

Bernardo Lanza Queiroz, Ph.D

Academic Editor

PLOS ONE

2. In the ethics statement in the manuscript and in the online submission form, please provide additional information about the patient data used in your retrospective study, including: a) whether all data were fully anonymized before you accessed them; b) the date range (month and year) during which the records were accessed.

3. We note that Figures 1-3  in your submission contain map images which may be copyrighted. All PLOS content is published under the Creative Commons Attribution License (CC BY 4.0), which means that the manuscript, images, and Supporting Information files will be freely available online, and any third party is permitted to access, download, copy, distribute, and use these materials in any way, even commercially, with proper attribution. For these reasons, we cannot publish previously copyrighted maps or satellite images created using proprietary data, such as Google software (Google Maps, Street View, and Earth). For more information, see our copyright guidelines: http://journals.plos.org/plosone/s/licenses-and-copyright.

(1) You may seek permission from the original copyright holder of Figures 1-3 to publish the content specifically under the CC BY 4.0 license. 

4. Please ensure you have discussed any potential limitations of your study in the Discussion.

5. Please include your tables as part of your main manuscript and remove the individual files. Please note that supplementary tables (should remain/ be uploaded) as separate "supporting information" files

Reviewers' comments:

Reviewer's Responses to Questions

**Comments to the Author**

1. Is the manuscript technically sound, and do the data support the conclusions?

Reviewer #1: Yes

Reviewer #2: No

2. Has the statistical analysis been performed appropriately and rigorously? 

Reviewer #1: Yes

Reviewer #2: I Don't Know

3. Have the authors made all data underlying the findings in their manuscript fully available?

Reviewer #1: Yes

Reviewer #2: Yes

4. Is the manuscript presented in an intelligible fashion and written in standard English?

Reviewer #1: Yes

Reviewer #2: Yes

5. Review Comments to the Author

Reviewer #1: The study showed excellent quality and the authors developed an important proposal for the epidemiological scenario of tuberculosis in Brazil.

The issue with immigrants needs to be discussed and studied more deeply, especially in Brazil. This priority group requires differentiated actions and strategies by health managers.

The authors clarified the Brazilian situation in recent years, but it could be interesting to discuss more deeply the limitations of access and the impact of social determinants on this group, a fact that can interfere in the control and spread of the disease.

I congratulate the group of authors for their clear and coherent writing as well as for the methodological process of spatial and temporal analysis that was conducted.

Reviewer #2: General comments

This descriptive study of TB among migrants is very relevant in a context of recent increased migration in Brazil. The paper is generally clear but I have some major concerns on the applicability of Getis-Ord Gi* analysis and time trends analysis to state that answer the study objective and describe that there is an increase number of TB cases among migrants in Brazil. By using spatial clusters, we are assuming that there is a continuum of cases within a given space, so the applicability of this method to analyse clusters within a subgroup (migrants) should be added as a limitation. Similarly, the limited applicability of time trends analysis in the context of a high influx of migrant populations to Brazil during the study period should be additionally discussed in the paper.

Other major points that may improve the quality of the paper include:

1. The title should be reconsidered – the results from this study can really bring light to if Brazil is managing to control the disease.

2. The overall language of the abstract is a little stigmatizing and differ from the overall language of paper, which was adequately written. The authors should try not to use sentences that bring old but already deconstructed myths about migrants (e.g., “Tuberculosis (TB) in migrants concerns health authorities worldwide, due to the potential for spread of the disease.”)

3. The methods are not properly referenced – all the analysis that are not very usual (e.g., Getis-Ord Gi analysis) should be adequately referenced.

4. In the two last paragraph of the results section, the authors describe the results for the association between TB incidence and social vulnerability. These results are relation to TB incidence overall or among migrants? This is not clear and, if it is overall, this analysis should be included as part of the objectives of the study as they differ from the overall objective of the paper. On the other hand, if this analysis refers to TB incidence among migrants, which denominator was used to calculate incidence?

5. The discussion do not included problems or suggestions on how to reduce social vulnerabilities of migrants so Brazil can diagnose and treat migrants timely. Also, the authors discussed their findings based on a limited number of references and do not adequately referenced the literature from the field - the authors should consider discussing the results in the light of other studies analysing TB characteristics in Brazil and other LMIC. Please see the lancet series of migration and health and other SR of TB among migrants.

6. The conclusion should be more straightforward and aligned to the overall study objective.

6. PLOS authors have the option to publish the peer review history of their article (what does this mean?). If published, this will include your full peer review and any attached files.

Reviewer #1: **Yes: **SHIRLEY VERONICA MELO ALMEIDA LIMA

Reviewer #2: No

---

## [Author Response · Author response to Decision Letter 0]

8 Apr 2021

Dear Editor, 

Many thanks for your reply and your reviewers' comments about our manuscript “PONE-D-20-40788 - Spatial clustering and temporal trend analysis of international migrants diagnosed with tuberculosis in Brazil”.

The comments were appropriate to qualify/improve the manuscript. We have forwarded a letter with the changes made in the manuscript, presenting our response for each comment from reviewers, as requested. Many thanks for the comments.

Kind regards,

Arcêncio et al.

Editor comments:

Comment: 1. Thank you for stating in your Funding Statement:

 [NO - he funders had no role in study design, data collection and analysis, decision to publish, or preparation of the manuscript.]. 

Response: Thank you for your comment. We inform that the author ELMN received received funding from the Pan American Health Organization (PAHO) [award number: 67278]. According to the journal's submission rules, this information was not included in the manuscript, only in the form of the submission system. We also inserted this statement in our cover letter, as directed.

Comment: 2. Please amend your authorship list in your manuscript file to include author Tatiana Ferraz de Araújo Alecrim

Response: Thank you for your observation and sorry for the mistake. This has already been corrected in the manuscript.

---

## [Decision Letter · Decision Letter 1]

27 Apr 2021

PONE-D-20-40788R1

Spatial clustering and temporal trend analysis of international migrants diagnosed with tuberculosis in Brazil

PLOS ONE

Dear Dr. Arcencio,

Thank you for submitting your manuscript to PLOS ONE. After careful consideration, we feel that it has merit but does not fully meet PLOS ONE’s publication criteria as it currently stands. Therefore, we invite you to submit a revised version of the manuscript that addresses the points raised during the review process.

Although the revised manuscript (R1) has been judged substantially improved in comparison with the original submission, there are important suggestions, comments and questions made by reviewer two still incompletely addressed. Please, make sure to completely address each question, comment and suggestion made by reviewer two to the R1 then resubmit it along with detailed explanations of how they were incorporated in the paper. Please, pay particular attention to the issue raised by that reviewer about tuberculosis among migrants in São Paulo.

We look forward to receiving your revised manuscript.

Kind regards,

Albert Schriefer, M.D., Ph.D.

Academic Editor

PLOS ONE

Journal Requirements:

Reviewers' comments:

Reviewer's Responses to Questions

**Comments to the Author**

1. If the authors have adequately addressed your comments raised in a previous round of review and you feel that this manuscript is now acceptable for publication, you may indicate that here to bypass the “Comments to the Author” section, enter your conflict of interest statement in the “Confidential to Editor” section, and submit your "Accept" recommendation.

Reviewer #1: All comments have been addressed

Reviewer #2: (No Response)

2. Is the manuscript technically sound, and do the data support the conclusions?

Reviewer #1: Yes

Reviewer #2: Yes

3. Has the statistical analysis been performed appropriately and rigorously? 

Reviewer #1: Yes

Reviewer #2: Yes

4. Have the authors made all data underlying the findings in their manuscript fully available?

Reviewer #1: Yes

Reviewer #2: Yes

5. Is the manuscript presented in an intelligible fashion and written in standard English?

Reviewer #1: Yes

Reviewer #2: Yes

6. Review Comments to the Author

Reviewer #1: (No Response)

Reviewer #2: In general, the study improved substantially since the first version of the manuscript. The main concerns/suggestions are listed below:

Abstract

Please revise the sentence “The study revealed the impact of TB in migrants. Based on the findings, if TB surveillance is not intensified in these areas, there will likely be an increase in cases in adjacent regions, making the target defined by the End TB Strategy more difficult to achieve”. How can the paper really revealed the impact of TB in migrants if it is not possible to determine the incidence of TB among migrant populations? If TB cases are being diagnosed and treated, why would the authors expect an increase in cases in adjacent regions is expected?

Introduction

The sentence “Several studies have shown the impact of migration on the resurgence and spread of TB in Europe” is inaccurate. The references 2 and 6-11 do show that cases among migrants pose challenges to local programmes but not necessarily that transmission contributes to it – e.g., Hargreaves (2017) show that there is low proportion of cases due to transmission from migrants to the local community, and that this is restricted to some communities. I suggest to remove the word spread.

Discussion

Page 18, row 338 – It needs to be more clear that the authors are referring to cases in “temporal trends of TB in migrants as increasing”.

Page 21, rows 387-393 - The authors say that TB cases among migrants in São Paulo have been stationary since 2014 and “Another hypothesis is underreporting and/or underdiagnosis in this population from 2014 to 2017, since the city did not report any migrants diagnosed with TB during that period.”. Nevertheless, the study by Pescarini (2018) (ref 38) reports data on over 335 migrants from South-American countries in 2013/2014 in Central areas from the municipality of São Paulo. There was any change in the National or State TB registry that may have caused this change? Please clarify the differences found and how these changes could have impacted the analysis.

7. PLOS authors have the option to publish the peer review history of their article (what does this mean?). If published, this will include your full peer review and any attached files.

Reviewer #1: **Yes: **SHIRLEY VERÔNICA MELO ALMEIDA LIMA

Reviewer #2: No

---

## [Author Response · Author response to Decision Letter 1]

3 May 2021

Dear Editor, 

Many thanks for your reply and your reviewers' comments about our manuscript “PONE-D-20-40788 - Spatial clustering and temporal trend analysis of international migrants diagnosed with tuberculosis in Brazil”.

The comments were appropriate to qualify/improve the manuscript. We have forwarded a letter with the changes made in the manuscript, presenting our response for each comment from reviewers, as requested. Many thanks for the comments.

Kind regards,

Arcêncio et al.

Editor comments:

Comment: Please review your reference list to ensure that it is complete and correct. If you have cited papers that have been retracted, please include the rationale for doing so in the manuscript text, or remove these references and replace them with relevant current references. Any changes to the reference list should be mentioned in the rebuttal letter that accompanies your revised manuscript. If you need to cite a retracted article, indicate the article’s retracted status in the References list and also include a citation and full reference for the retraction notice.

Response: Thank you for your comment. All references have been revised.

Reviewer #2:

Comment: Abstract: Please revise the sentence “The study revealed the impact of TB in migrants. Based on the findings, if TB surveillance is not intensified in these areas, there will likely be an increase in cases in adjacent regions, making the target defined by the End TB Strategy more difficult to achieve”. How can the paper really revealed the impact of TB in migrants if it is not possible to determine the incidence of TB among migrant populations? If TB cases are being diagnosed and treated, why would the authors expect an increase in cases in adjacent regions is expected?

Response: Thank you for your comment and we agree with your observation so that we rewrite the conclusion of the abstract.

Comment: Introduction: The sentence “Several studies have shown the impact of migration on the resurgence and spread of TB in Europe” is inaccurate. The references 2 and 6-11 do show that cases among migrants pose challenges to local programmes but not necessarily that transmission contributes to it – e.g., Hargreaves (2017) show that there is low proportion of cases due to transmission from migrants to the local community, and that this is restricted to some communities. I suggest to remove the word spread.

Response: Thank you for your comment, we agree to observe it and as suggested, we remove the word "spread" from the sentence.

Comment: Discussion: Page 18, row 338 – It needs to be more clear that the authors are referring to cases in “temporal trends of TB in migrants as increasing”.

Response: Thank you for your comment. We rewrote the sentence to make it clearer, as suggested.

Comment: Page 21, rows 387-393 - The authors say that TB cases among migrants in São Paulo have been stationary since 2014 and “Another hypothesis is underreporting and/or underdiagnosis in this population from 2014 to 2017, since the city did not report any migrants diagnosed with TB during that period.”. Nevertheless, the study by Pescarini (2018) (ref 38) reports data on over 335 migrants from South-American countries in 2013/2014 in Central areas from the municipality of São Paulo. There was any change in the National or State TB registry that may have caused this change? Please clarify the differences found and how these changes could have impacted the analysis.

Response: This is an excellent observation that also made us thoughtful.

The data from our study were obtained through SINAN, which is the official notification system in Brazil, as well as data from the study by Pescarini (2018), which also used data provided by the laboratories of the Adolfo Lutz Institute (IAL).

The SINAN database should be the same, both for us and for Pescarini. We were unable to say about the database received by Pescarini, but in our databank there are approximately 45 thousand cases of TB reported in São Paulo only during the period under study (2014 to 2019), with 7 thousand cases only in 2014, but not there were TB cases registered as being in migrants between the years 2014 and 2017 in São Paulo. It is worth mentioning that in Pescarini's study, he selected only the central region of São Paulo and here we are talking about the municipality as a whole, so the numbers I quote here are different from the numbers presented in Pescarini's study.

Considering that theoretically the SINAN bank is the same for everyone who has access, we can conclude that the cases of migrants considered by Pescarini in his study were obtained through the IAL laboratory exam database, so that when the exam was carried out , the person reported being a migrant, but that information was not entered in SINAN. Knowing these people, it is possible to cross-check the databases to obtain the other variables that Pescarini used in his study.

Studies using secondary databases are subject to these limitations regarding unreported data, typing errors and other issues, as discussed in the manuscript, so that the analyzes made and presented were carried out with methodological rigor in order to reduce possible biases, however we work with the data provided to us and we are able to answer only for them.

Sorry if we could not clarify your question with further information. We inform that we inserted a paragraph in the discussion session referring to the possibility of crossing databases from different sources to minimize the bias already mentioned.

---

## [Decision Letter · Decision Letter 2]

21 May 2021

Spatial clustering and temporal trend analysis of international migrants diagnosed with tuberculosis in Brazil

PONE-D-20-40788R2

Dear Dr. Arcencio,

We’re pleased to inform you that your manuscript has been judged scientifically suitable for publication and will be formally accepted for publication once it meets all outstanding technical requirements.

Kind regards,

Albert Schriefer, M.D., Ph.D.

Academic Editor

PLOS ONE

Additional Editor Comments (optional):

Reviewers' comments:

Reviewer's Responses to Questions

**Comments to the Author**

1. If the authors have adequately addressed your comments raised in a previous round of review and you feel that this manuscript is now acceptable for publication, you may indicate that here to bypass the “Comments to the Author” section, enter your conflict of interest statement in the “Confidential to Editor” section, and submit your "Accept" recommendation.

Reviewer #2: All comments have been addressed

2. Is the manuscript technically sound, and do the data support the conclusions?

Reviewer #2: Yes

3. Has the statistical analysis been performed appropriately and rigorously? 

Reviewer #2: Yes

4. Have the authors made all data underlying the findings in their manuscript fully available?

Reviewer #2: Yes

5. Is the manuscript presented in an intelligible fashion and written in standard English?

Reviewer #2: Yes

6. Review Comments to the Author

Reviewer #2: The authors have addressed the main points raised in the previous version. However, there are two points I would like to raise that might reflect my personal opinion (but that might still be relevant):

1. The study justification states that there is a lack of studies on the potential resurgence of TB in Brazil due to migration (and that is a big gap in the literature), which I disagree with as Brazil is still a high burden TB country with approximately 30 thousand TB cases a year. I think the study is very important to guide actions as this population is very vulnerable, but it is unlikely that those cases make TB incidence to increase substantially.

2. Regarding the interpretation of the results, there is still some room for potential discriminatory language (e.g., abstract: “ensuring that these cases concluded their treatment and avoiding that they could spread the disease to the other regions or scenarios.”). In the scenario of xenophobia all over the world, including in Brazil, I believe we should be careful saying that migrants could spread a disease that is not even controlled in the country.

7. PLOS authors have the option to publish the peer review history of their article (what does this mean?). If published, this will include your full peer review and any attached files.

Reviewer #2: **Yes: **Julia Pescarini

---

## [Editor Report · Acceptance letter]

31 May 2021

PONE-D-20-40788R2 

Spatial clustering and temporal trend analysis of international migrants diagnosed with tuberculosis in Brazil 

Dear Dr. Arcêncio:

I'm pleased to inform you that your manuscript has been deemed suitable for publication in PLOS ONE. Congratulations! Your manuscript is now with our production department. 

Kind regards, 

on behalf of

Dr. Albert Schriefer 

Academic Editor

PLOS ONE